# 🦉 Tracing the Hidden: Segment Anything in Camouflaged Videos via Prompt-Free Multimodal LLM Guidance

## Abstract

Camouflaged object segmentation in videos faces inherent challenges due to the targets' indistinguishable appearance and irregular motion patterns. While Segment Anything Model 2 (SAM2) provides a flexible framework for prompt-driven segmentation, it heavily relies on handcrafted or external prompts, limiting its potential in complex, real-world scenarios. To address the issue, we present Camo-Tracer, a prompt-free yet prompt-rich framework that leverages multimodal large language models (MLLMs) to generate diverse and informative prompts, i.e., point, mask and text prompts, to guide SAM2 without any human intervention. We introduce two key components: (1) a Semantic-Guided Adapter that aligns CLIP and SAM2 representations via cross-attention, injecting rich semantic context into high-resolution visual features; and (2) a Semantic-Aware Prompter that transforms semantic response maps into coarse masks and Gumbel-Softmax-based sampling points, which allows end-to-end differentiable optimization. Meanwhile, LLM outputs text tokens to derive implicit text prompts that encode rich visual-language priors. These prompts collaboratively guide the SAM2 mask decoder in a self-adaptive manner. Further, we devise a memory-guided bi-directional keyframe selection strategy to enhance temporal context propagation and prompt reliability across video frames. Extensive experiments on VCOS benchmarks, MoCA-Mask and CAD datasets, demonstrate that CamoTracer achieves new state-of-the-art performance, strong generalization ability, and robust prompt adaptation, outperforming previous approaches by a significant margin. Our results highlight the potential of self-prompted segmentation empowered by multimodal understanding, bringing SAM2 one step closer to human-like perception in camouflaged scenes.

## 1 Introduction

Camouflaged object segmentation (COS)[1] is a crucial and challenging task in computer vision, aiming to segment objects that blend seamlessly into their surroundings. The inherent ambiguity in object appearance makes COS particularly difficult, as camouflaged targets often exhibit low contrast against the background and lack clear semantic boundaries. Recent advances in this field have deepened the insights into camouflage patterns and enabled practical applications in various fields, e.g., medical image analysis (Bao et al., 2024; Huang et al., 2024a; Zhang et al., 2024b; Wolleb et al., 2022; Zhao et al., 2021), industrial defect detection (Cao et al., 2023; Roth et al., 2022; Liu et al., 2021), and wildlife conservation (Lidbetter, 2020).

Extending COS to the video domain, video COS (VCOS) introduces additional challenges that are unique to temporal modeling. These challenges include not only visual ambiguity caused by appearance similarity between objects and backgrounds, but also prediction instability arising from scene dynamics, such as occlusion, sudden object emergence, and motion blur. While temporal information can reveal subtle appearance changes, accurately modeling motion in camouflaged scenarios remains non-trivial. Moreover, objects may remain motionless or be visually indistinct, making both appearance- and motion-based detection inherently unreliable.

---

[1]Also termed as camouflaged object detection (COD). Throughout, we use COS and COD interchangeably.

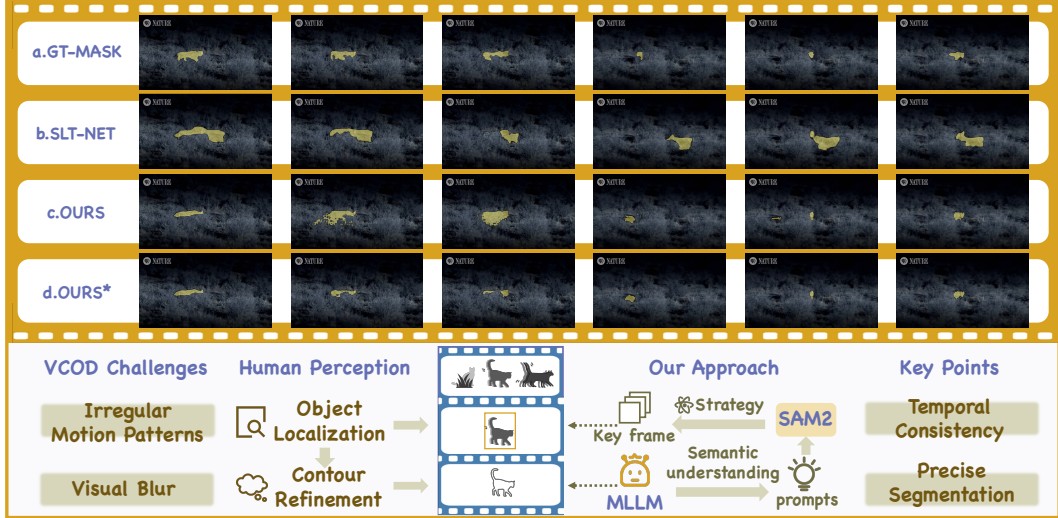

Figure 1: (a) Comparison of mask predictions between ground truth (GT), SLT-Net (Cheng et al., 2022a), baseline (LISA (Lai et al., 2024)) and CamoTracer (Ours). (b) Illustration of challenges and our motivation.

Existing VCOS methods (Bideau & Learned-Miller, 2016a; Lamdouar et al., 2020; Cheng et al., 2022a; Yu et al., 2024b; Hui et al., 2024a) mainly leverage optical flow or temporal correlation for motion-aware segmentation. However, (i) motion estimation can be erroneous in low-contrast scenes, leading to error accumulation in downstream predictions; and (ii) many methods rely heavily on limited annotated data, leading to poor generalization across diverse scenes. These limitations highlight the need for a more robust and universal solution to handle dynamic and ambiguous camouflage patterns.

Segment Anything Model 2 (SAM2) (Ravi et al., 2024) represents a major advancement in video object segmentation. As a prompt-driven foundation model, SAM2 demonstrates strong generalization across domains. However, SAM2's reliance on external prompts makes it difficult to adapt to COS, where providing reliable prompts is particularly challenging due to the lack of distinguishable visual cues. Recent works (Hui et al., 2024b; Meeran et al., 2024; Zhang et al., 2025a) attempt to integrate appearance-motion heuristics or self-prompting strategies to enhance SAM2 for VCOS, but these methods are still limited by the ambiguity of appearance and the noise in motion estimation. In such camouflaged settings, generating high-quality prompts without human intervention remains a key bottleneck.

To address this, we draw inspiration from human perception, as illustrated in Fig. 1. When faced with camouflaged objects, humans tend to rely on subtle motion cues for initial identification. Once the object is recognized, the memory of its features facilitates subsequent detection, even under occlusion or static conditions. Inspired by the role of semantic reasoning and temporal memory in human vision, we aim to endow SAM2 with human-like perception by integrating multimodal large language models (MLLMs). MLLMs possess powerful semantic reasoning capabilities and can synthesize visual-language priors to detect subtle targets. We leverage MLLMs to generate diverse multimodal prompts, thereby replacing human intervention with adaptive, semantics-rich guidance.

We present CamoTracer, a prompt-free yet prompt-rich framework that combines MLLMs and SAM2 for robust VCOS. Specifically, we introduce two key components: (i) the Semantic-Guided Adapter (SGA), which injects semantic context from MLLMs into SAM2 via cross-attention, aligning visual and semantic representations; and (ii) the Semantic-Aware Prompter (SAP), which converts MLLM outputs into diverse and complementary prompts (text, point, mask) to enrich the prompt space and enhance segmentation quality.

To further strengthen temporal consistency and mitigate challenges such as motion variability and occlusions, we propose a memory-guided Bi-directional Keyframe Selection (Bi-KFS) strategy. This strategy utilizes bidirectional inference consistency and mask prediction confidence to select reliable keyframes as memory anchors, thereby stabilizing the segmentation process across frames. By

ensuring that contextual information from keyframes propagates effectively to subsequent frames, our approach reduces the impact of drift and maintains consistent segmentation throughout the video sequence.

Our contributions are summarized as follows:

- We propose **CamoTracer**, the first MLLM-enhanced VCOS framework that enables fully automated prompt generation for SAM2, removing the need for human intervention and significantly enhancing segmentation quality in camouflage scenarios.

- To address the challenge of visual ambiguity, we introduce the **Semantic-Guided Adapter** and **Semantic-Aware Prompter**, effectively aligning visual and semantic representations and providing robust prompt guidance, endowing SAM2 with human-like perception.

- To mitigate the instability caused by inter-frame discontinuities and irregular motion, we design a **Bi-directional Keyframe Selection** strategy, which identifies optimal memory anchors to enhance long-term temporal propagation and prediction consistency.

- Extensive experiments on MoCA-Mask and CAD2016 benchmarks demonstrate the superiority of CamoTracer, surpassing the previous state-of-the-art method CamoSAM2 by **+9.4%** and **+22.3%** mIoU, respectively.

## 2 RELATED WORK

### 2.1 VIDEO CAMOUFLAGED OBJECT SEGMENTATION

VCOS (Xiao et al., 2024; Bi et al., 2021) presents unique challenges, primarily due to the need to utilize motion cues to differentiate targets with indistinguishable appearances. Traditional VCOD approaches rely on optical flow (Bideau & Learned-Miller, 2016a; Lamdouar et al., 2020) to capture motion cues between video frames but suffered from accumulated mask errors in dynamic scenes. SLT-Net (Cheng et al., 2022a) addresses this by proposing a two-stage framework that models both short- and long-term temporal consistency. TMNet (Yu et al., 2024b) enhances the segmentation accuracy by using motion-guided features extracted via learnable token selection, while IMEX (Hui et al., 2024a) integrates both implicit and explicit motion learning through cross-scale fusion.

The limited availability of training data often restricts the generalization capabilities of these models, prompting the need for more generalizable solutions. In response, TSP-SAM (Hui et al., 2024b) introduces temporal-spatial prompt learning within the Segment Anything Model (SAM), enabling the automatic generation of prompts based on motion cues. SAM-PM (Meeran et al., 2024) builds on SAM by incorporating a propagation module to enforce temporal consistency in segmentation. CamoSAM2 (Zhang et al., 2025a) enhances SAM2's performance for VCOD tasks by introducing a motion-appearance prompt inducer and an adaptive multi-prompt refinement strategy.

Existing methods rely on prompts based on appearance and motion, overlooking the semantic requirements inherent in camouflaged object detection tasks. In contrast, our approach incorporates MLLMs, effectively addressing the challenges of both visual and semantic ambiguities.

### 2.2 SEGMENT ANYTHING MODEL 2

SAM2 (Ravi et al., 2024) represents a significant advancement over its predecessor, SAM (Kirillov et al., 2023), by enabling a universal vision segmentation model that spans both image and video tasks. While SAM was confined to image segmentation, SAM2 extends its capabilities by incorporating a memory structure, allowing it to handle temporal dependencies. This addition has enabled SAM2 to achieve a remarkable leap in the domain of natural video segmentation, particularly in its zero-shot capabilities.

Despite its success, SAM2's performance in specialized fields remains limited. To address this, several studies have tailored SAM2 for specific domains, such as medical image segmentation (Yu et al., 2024a; Chen et al., 2024; Mansoori et al., 2024; Zhu et al., 2024), video object tracking (Zhang et al., 2024a; Stanczyk & Bremond, 2024), point cloud segmentation (Guo et al., 2024), and video camouflaged object segmentation (Zhang et al., 2025a; Zhou et al., 2024; Tang & Li, 2024), demonstrating its versatility in these areas. However, it is important to note that SAM2's performance

heavily relies on the quality of the provided prompts. Current methods like medicineSAM (Zhu et al., 2024) and SAM-PM (Meeran et al., 2024) still depend significantly on handcrafted or external prompts, which restrict the model's potential in more complex, real-world scenarios.

In contrast, our approach focuses on automating the generation of diverse and informative prompts for SAM2, without any human intervention. By leveraging MLLMs, the prompts we generate are not only rich in appearance and motion cues but also integrate semantic understanding into SAM2.

### 2.3 MULTIMODAL LARGE LANGUAGE MODEL

MLLMs have recently recorded striking breakthroughs on vision-language tasks. Some studies (Alayrac et al., 2022; Li et al., 2023a;b; Liu et al., 2023; Zhu et al., 2023; Ye et al., 2023; Lei et al., 2025) have made groundbreaking progress. BLIP-2 (Li et al., 2023b) and Mplug-Owl (Ye et al., 2023) use a two-stage design, combining image embeddings with text tokens for zero-shot transfer via a frozen LLM. In contrast, LLaVA (Liu et al., 2023) and MiniGPT-4 (Zhu et al., 2023) explicitly project visual features into the language space and apply visual-instruction tuning for interactive instruction following.

Building on such expressive representations, Wang et al. (2023); Chen et al. (2023); Zhang et al. have moved from holistic description to explicit grounding. At the same time, MLLMs have proved adaptable to diverse visual downstream tasks such as multimodal generation (Ye et al., 2024; Dong et al., 2023), object detection (Jiao et al., 2024), and image segmentation (Lai et al., 2024; Ren et al., 2024b; Tang et al., 2025); in particular, LISA (Lai et al., 2024) decodes the hidden state of a dedicated <SEG> token into open-set masks, whereas PixelLM (Ren et al., 2024b) internalises a segmentation codebook and pixel decoder to produce multi-object masks.

Building on image-level progress, recent work extends MLLMs to video: LITA (Huang et al., 2024b) uses relative time tokens for temporal localization; TimeChat (Ren et al., 2024a) combines a timestamp-aware encoder with a sliding Q-Former; and Momentor (Qian et al., 2024) learns continuous temporal embeddings from Moment-10M. However, these methods remain instruction-driven, differing from our target of video camouflage segmentation. To bridge this gap, we introduce the first MLLM-based VCOS framework that is prompt-free yet retains prompt-rich features.

## 3 METHOD

Our proposed CamoTracer pioneers the integration of MLLMs with SAM2 for video camouflaged object segmentation. By leveraging the rich semantic priors from MLLMs, CamoTracer automatically generates informative prompts that guide the promptable segmentation model SAM2, setting a new precedent for semantic-driven, video-based segmentation in camouflaged scenarios.

### 3.1 ARCHITECTURE

Fig. 2 illustrates the overall architecture of CamoTracer, which integrates the promptable segmentation model SAM2 with the MLLM. Given a video clip $\mathbf{x}_{\text{video}} = \{\mathbf{x}_{\text{img}}^{(1)}, \mathbf{x}_{\text{img}}^{(2)}, \ldots, \mathbf{x}_{\text{img}}^{(T)}\}$, where $\mathbf{x}_{\text{img}}^{(t)}$ represents the individual image frame at time step $t$, the model processes the frames sequentially, utilizing contextual information and memory from previous frames to generate segmentation masks.

Each input frame $\mathbf{x}_{\text{img}}^{(t)}$ is combined with a fixed text instruction, formatted as: "`<IMAGE> Please segment the camouflaged object in this image.`", where `<IMAGE>` serves as a placeholder for the image patch tokens. To enable the LLM to assist in segmentation, we extend the LLM's vocabulary with a special token `<SEG>`, following previous work (Lai et al., 2024). The CLIP image encoder processes the input image and encodes it into visual tokens, which are passed to the LLM. The LLM then generates a text-based response $\hat{\mathbf{y}}_{\text{txt}}$, formulated as:

$$\hat{\mathbf{y}}_{\text{txt}} = \mathcal{F}(\mathbf{x}_{\text{img}}^{(t)}, \mathbf{x}_{\text{txt}}). \tag{1}$$

The embedding corresponding to the `<SEG>` token, $\tilde{\mathbf{h}}_{\text{seg}}$, is extracted from the last layer of the LLM and passed through a projection layer $\gamma$ to obtain a feature embedding $\mathbf{h}_{\text{seg}}$, which serves as a

Figure 2: Overall architecture of CamoTracer. Our framework integrates MLLM's semantic understanding with SAM2's promptable segmentation capabilities.

text prompt for SAM2's mask decoder. Simultaneously, the SAM2 image encoder, $\mathcal{F}_{\text{sam2}}$, extracts multi-scale visual features $\mathbf{f}_{\text{sam2}}$ from each frame $\mathbf{x}_{\text{img}}^{(t)}$:

$$\mathbf{h}_{\text{seg}} = \gamma(\tilde{\mathbf{h}}_{\text{seg}}), \quad \mathbf{f}_{\text{sam2}} = \mathcal{F}_{\text{sam2}}(\mathbf{x}_{\text{img}}^{(t)}). \tag{2}$$

The SAM2 captures fine-grained spatial details, while the CLIP encoder provides high-level semantic representations. To align these two feature spaces, we introduce the Semantic-Guided Adapter (SGA), which injects semantic priors into the visual stream. This module will be introduced in Section 3.2.

While LLM generates text token embeddings as prompts for SAM2's mask decoder, these prompts lack explicit spatial cues, which are crucial for accurate segmentation. To address this limitation, we propose the Semantic-Aware Prompter (SAP), which derives semantic responses from the MLLM to generate both mask and point prompts, further improving segmentation performance. Details of SAP are provided in Section 3.3.

Our model employs a streaming processing and memory-based prompting mechanism, which enhances segmentation stability by leveraging context from previous frames. However, for camouflaged objects that are partially visible or gradually emerge across frames, earlier frame predictions may propagate noise, leading to error accumulation. To mitigate this issue, we introduce a motion-guided bidirectional keyframe selection strategy to enhance temporal context propagation and ensure segmentation consistency. This strategy will be elaborated in Section 3.5.

## 3.2 SEMANTIC-GUIDED ADAPTER

Camouflaged objects often blend seamlessly into the background, making it difficult to distinguish targets from distractors using visual cues alone. This calls for external semantic guidance to disambiguate object regions. While the SAM2 encoder provides rich low- and mid-level visual features, it lacks the high-level semantic grounding required to accurately localize camouflaged objects. To address this limitation, we introduce a lightweight Semantic-Guided Adapter (SGA) that injects high-level semantic priors into the visual feature stream, enhancing the semantic discriminability of visual representations.

Specifically, SGA takes the visual features from SAM2 $\mathbf{f}_{\text{sam2}}$ as queries and applies a lightweight cross-attention mechanism conditioned on CLIP features $\mathbf{f}_{\text{clip}}$, followed by a layer normalization:

$$\mathbf{f}_{\text{adapted}} = \text{LayerNorm}(\text{CrossAttn}(\mathbf{f}_{\text{sam2}}, \mathbf{f}_{\text{clip}})). \tag{3}$$

The adapted features are fused back with the original SAM2 features to produce semantically enriched visual representations:

$$\mathbf{f}_{\text{fused}} = \mathbf{f}_{\text{sam2}} + \mathbf{f}_{\text{adapted}}. \tag{4}$$

This fusion integrates fine-grained spatial details with global semantic context, allowing the model to attend to semantically meaningful regions even under weak visual contrast. The adapter is trained end-to-end and introduces minimal additional parameters.

### 3.3 SEMANTIC-AWARE PROMPTER

Although text prompts provide high-level semantic guidance, they lack explicit spatial cues required for accurate localization. To compensate for this, we propose the Semantic-Aware Prompter (SAP), which extracts semantic response maps from aligned vision-language features and converts them into mask prompts. We first obtain the enhanced features $\mathbf{f}'_{\text{clip}}$ via a Global-Local Fusion (GLF) module that integrates early- and final-layer features from CLIP, capturing both local details and global semantics. This can be formulated as:

$$
\begin{aligned}
\mathbf{f}'_{\text{clip}} &= \Psi(\Phi(\mathbf{f}_{\text{clip}})) + \Psi(\Phi(\mathbf{f}_{\text{early}})), \\
\Phi(\cdot) &= \text{GELU}(\text{LayerNorm}(\text{TransConv}(\cdot))), \\
\Psi(\cdot) &= \text{GELU}(\text{TransConv}(\cdot)),
\end{aligned}
\tag{5}
$$

where TransConv denotes transposed convolution, and GELU is the GELU activation function. Then, we calculate the text-pixel response maps through the inner product, which are reshaped to obtain mask predictions $\hat{\mathbf{n}}$ of the target with low-resolution:

$$
\hat{\mathbf{n}} = \mathbf{h}_{\text{seg}} \cdot \mathbf{f}'_{\text{clip}},
\tag{6}
$$

where $\hat{\mathbf{n}} \in \mathbb{R}^{H \times W}$ highlights potential target regions. This map is upsampled to a higher resolution $\hat{\mathbf{n}}'$, and encoded into a mask prompt embedding $\mathbf{f}_m$ using the SAM2 prompt encoder $\mathcal{F}_{\text{enc}}$:

$$
\mathbf{f}_m = \mathcal{F}_{\text{enc}}(\hat{\mathbf{n}}').
\tag{7}
$$

To further enrich the spatial prompt, we extract a representative point from $\hat{\mathbf{n}}$ that indicates the highest activation location. However, the conventional `argmax` operation is non-differentiable and blocks gradient flow. To overcome this, we employ the Gumbel-Softmax trick (Jang et al., 2016) to produce a differentiable one-hot spatial map $\mathcal{M} \in \mathbb{R}^{H \times W}$:

$$
\mathcal{M} = \text{Gumbel-Softmax}(\hat{\mathbf{n}}),
\tag{8}
$$

where $\mathcal{M}$ softly approximates hard point selection in a gradient-friendly manner. Since the SAM2 prompt encoder is frozen during training, we precompute the position embeddings $E \in \mathbb{R}^{H \times W \times D}$ for all spatial locations as a lookup table. The final point embedding $\mathbf{h}_p \in \mathbb{R}^D$ is then retrieved via a weighted sum over $\mathcal{M}$:

$$
\mathbf{h}_p = \sum_{i=1}^{H \times W} \mathcal{M}_i \cdot E_i.
\tag{9}
$$

To enhance temporal consistency, the fused embedding $\mathbf{f}_{\text{fuse}}$ and the mask prediction $\hat{\mathbf{m}}$ are fed into SAM2's memory encoder to produce the memory embedding $\mathbf{f}_{\text{mem}}$, which integrates information from the current and previous keyframes, guiding the segmentation of the current frame:

$$
\mathbf{f}_{\text{mem}} = \mathcal{F}_{\text{mem}}(\mathbf{f}_{\text{fuse}}, \hat{\mathbf{m}}).
\tag{10}
$$

This memory feature $\mathbf{f}_{\text{mem}}$ is then used as an additional input to the SAM2 mask decoder, alongside the text prompt $\mathbf{h}_{\text{seg}}$, the mask prompt $\mathbf{f}_m$, and the point prompt $\mathbf{h}_p$:

$$
\hat{\mathbf{m}} = \mathcal{F}_{\text{dec}}(\mathbf{h}_{\text{seg}}, \mathbf{f}_m, \mathbf{h}_p, \mathbf{f}_{\text{mem}}).
\tag{11}
$$

### 3.4 TRAINING

**Training Objectives.** Our model is trained end-to-end with a multi-task loss that jointly optimizes text generation and segmentation performance. The overall loss $\mathcal{L}$ is a weighted sum of an autoregressive cross-entropy loss for text generation $\mathcal{L}_{txt}$ as well as segmentation losses for mask prediction $\mathcal{L}_{mask}$ and coarse mask prediction $\mathcal{L}_{coarse}$, with corresponding loss weights $\lambda_{txt}$, $\lambda_{mask}$, and $\lambda_{coarse}$:

$$
\mathcal{L} = \lambda_{\text{txt}}\mathcal{L}_{\text{txt}} + \lambda_{\text{mask}}\mathcal{L}_{\text{mask}} + \lambda_{\text{coarse}}\mathcal{L}_{\text{coarse}}.
\tag{12}
$$

The segmentation losses $\mathcal{L}_{\text{mask}}$ and $\mathcal{L}_{\text{coarse}}$ are computed as a combination of binary cross-entropy (BCE) and DICE loss, with weights $\lambda_{\text{bce}}$ and $\lambda_{\text{dice}}$, respectively. Given the ground-truth targets $(\mathbf{y}_{\text{txt}}, \mathbf{m}, \mathbf{n})$ and predictions $(\hat{\mathbf{y}}_{\text{txt}}, \hat{\mathbf{m}}, \hat{\mathbf{n}})$, where $\mathbf{m}$ and $\mathbf{n}$ denote the final and coarse masks, the losses are defined as:

$$
\begin{aligned}
\mathcal{L}_{\text{txt}} &= \text{CE}(\hat{\mathbf{y}}_{\text{txt}}, \mathbf{y}_{\text{txt}}), \\
\mathcal{L}_{\text{mask}} &= \lambda_{\text{bce}} \, \text{BCE}(\hat{\mathbf{m}}, \mathbf{m}) + \lambda_{\text{dice}} \, \text{DICE}(\hat{\mathbf{m}}, \mathbf{m}), \\
\mathcal{L}_{\text{coarse}} &= \lambda_{\text{bce}} \, \text{BCE}(\hat{\mathbf{n}}, \mathbf{n}) + \lambda_{\text{dice}} \, \text{DICE}(\hat{\mathbf{n}}, \mathbf{n}).
\end{aligned}
\tag{13}
$$

Table 1: Comparison with SOTA methods on the MoCA-Mask and CAD2016 datasets. The best and the second-best results are **bolded** and underlined, respectively.

| Method | Pub./Year | Input | MoCA-Mask | | | | | | CAD2016 | | | | | |
|---|---|---|---|---|---|---|---|---|---|---|---|---|---|---|
| | | | $S_\alpha \uparrow$ | $F_\beta^w \uparrow$ | $E_\phi \uparrow$ | $\mathcal{M} \downarrow$ | $mDice \uparrow$ | $mIoU \uparrow$ | $S_\alpha \uparrow$ | $F_\beta^w \uparrow$ | $E_\phi \uparrow$ | $\mathcal{M} \downarrow$ | $mDice \uparrow$ | $mIoU \uparrow$ |
| SINet (Fan et al., 2020) | CVPR$_{2020}$ | Image | 0.574 | 0.185 | 0.655 | 0.030 | 0.221 | 0.156 | 0.601 | 0.204 | 0.589 | 0.089 | 0.289 | 0.209 |
| SINet-v2 (Fan et al., 2021a) | TPAMI$_{2021}$ | Image | 0.571 | 0.175 | 0.608 | 0.035 | 0.211 | 0.153 | 0.544 | 0.126 | 0.546 | 0.049 | 0.170 | 0.110 |
| ZoomNet (Pang et al., 2022) | CVPR$_{2022}$ | Image | 0.582 | 0.211 | 0.536 | 0.033 | 0.224 | 0.167 | 0.587 | 0.225 | 0.594 | 0.063 | 0.246 | 0.166 |
| BGNet (Sun et al., 2022) | IJCAI$_{2022}$ | Image | 0.590 | 0.203 | 0.647 | 0.023 | 0.225 | 0.167 | 0.607 | 0.203 | 0.666 | 0.089 | 0.345 | 0.256 |
| FEDERNet (He et al., 2023) | CVPR$_{2023}$ | Image | 0.555 | 0.158 | 0.542 | 0.049 | 0.192 | 0.132 | 0.607 | 0.246 | 0.725 | 0.061 | 0.361 | 0.257 |
| FSPNet (Huang et al., 2023) | CVPR$_{2023}$ | Image | 0.594 | 0.182 | 0.608 | 0.044 | 0.238 | 0.167 | 0.539 | 0.220 | 0.553 | 0.145 | 0.309 | 0.212 |
| PUENet (Zhang et al., 2023) | TIP$_{2023}$ | Image | 0.594 | 0.204 | 0.619 | 0.037 | 0.302 | 0.212 | 0.673 | 0.427 | 0.803 | 0.034 | 0.499 | 0.389 |
| RCRNet (Yan et al., 2019) | ICCV$_{2019}$ | Video | 0.597 | 0.174 | 0.583 | 0.025 | 0.194 | 0.137 | 0.627 | 0.287 | 0.666 | 0.048 | 0.309 | 0.229 |
| PNS-Net (Ji et al., 2021) | MICCAI$_{2021}$ | Video | 0.576 | 0.134 | 0.562 | 0.038 | 0.189 | 0.133 | 0.678 | 0.369 | 0.720 | 0.043 | 0.409 | 0.308 |
| MG (Yang et al., 2021) | ICCV$_{2021}$ | Video | 0.547 | 0.165 | 0.537 | 0.095 | 0.197 | 0.141 | 0.484 | 0.314 | 0.558 | 0.370 | 0.351 | 0.260 |
| SLT-Net (Cheng et al., 2022a) | CVPR$_{2022}$ | Video | 0.656 | 0.357 | 0.785 | 0.021 | 0.387 | 0.310 | 0.679 | 0.420 | 0.805 | 0.033 | 0.445 | 0.342 |
| ZoomNeXt (Pang et al., 2024) | TPAMI$_{2024}$ | Video | 0.734 | 0.476 | 0.497 | 0.010 | 0.497 | 0.422 | 0.757 | 0.593 | 0.865 | 0.020 | 0.599 | 0.510 |
| TMNet (Yu et al., 2024b) | ICASSP$_{2024}$ | Video | 0.740 | 0.485 | 0.735 | 0.008 | 0.503 | 0.417 | - | - | - | - | - | - |
| IMEX (Hui et al., 2024a) | TMM$_{2024}$ | Video | 0.661 | 0.371 | 0.778 | 0.020 | 0.409 | 0.319 | 0.684 | 0.452 | 0.813 | 0.033 | 0.469 | 0.370 |
| TSP-SAM (Hui et al., 2024b) | CVPR$_{2024}$ | Video | 0.689 | 0.444 | 0.808 | 0.008 | 0.458 | 0.388 | 0.704 | 0.524 | 0.912 | 0.028 | 0.543 | 0.438 |
| SAM-PM (Meeran et al., 2024) | CVPRW$_{2024}$ | Video | 0.728 | 0.567 | 0.813 | 0.009 | 0.594 | 0.502 | 0.729 | 0.602 | 0.746 | 0.018 | 0.594 | 0.493 |
| EMIP (Zhang et al., 2025b) | TIP$_{2025}$ | Video | 0.675 | 0.381 | - | 0.015 | 0.426 | 0.333 | 0.719 | 0.514 | - | 0.028 | 0.536 | 0.425 |
| Vcamba (Li et al., 2025) | Arxiv$_{2025}$ | Video | 0.684 | 0.382 | 0.804 | 0.010 | 0.459 | 0.369 | 0.729 | 0.573 | 0.842 | 0.034 | 0.634 | 0.509 |
| CamoSAM2 (Zhang et al., 2025a) | Arxiv$_{2025}$ | Video | 0.765 | 0.607 | 0.848 | 0.007 | 0.620 | 0.542 | 0.774 | 0.652 | 0.852 | 0.018 | 0.647 | 0.543 |
| CamoTracer | Ours | Video | **0.800** | **0.665** | **0.878** | **0.006** | **0.674** | **0.593** | **0.830** | **0.745** | **0.900** | **0.014** | **0.750** | **0.664** |

**Training Strategies.** To preserve the rich pre-trained knowledge embedded in the LLM, we adopt LoRA (Hu et al., 2022) for parameter-efficient fine-tuning, and freeze all components of SAM2 except for its mask decoder $\mathcal{F}_{dec}$. Additionally, the LLM token embeddings, the LLM head, the projection layer $\gamma$, the SGA and SAP modules are trainable. This strategy maintains the robustness of the pre-trained models while adapting them effectively to the VCOS task. To further enhance generalization and reduce the model's over-reliance on specific prompt types, we apply random dropout on point, mask, text or memory prompts during training.

### 3.5 Memory-Guided Bi-Directional Keyframe Selection

To enhance segmentation stability and reduce temporal error accumulation in camouflaged videos, we introduce a memory-guided bi-directional keyframe selection strategy. Given a sequence of $T$ frames, we apply SAM2 in both forward and backward directions to obtain predicted masks $\{M_t^{fwd}\}_{t=1}^T$ and $\{M_t^{bwd}\}_{t=1}^T$. For each frame $t$, we compute a forward-backward consistency score:

$$S_t = \text{IoU}(M_t^{fwd}, M_t^{bwd}), \tag{14}$$

which reflects the agreement between bi-directional predictions. We rank frames by $S_t$ and select the top-$K$ (empirically $K = 3$) pairs. For each pair, the frame with the higher predicted mask confidence (IoU) is chosen as a keyframe. Additionally, any frame with a mask confidence exceeding a threshold (0.95) is also selected. A frame is considered a keyframe if it satisfies either criterion.

Finally, we re-run inference by first processing keyframes and then the remaining frames, using keyframes as memory anchors to guide segmentation. The detailed algorithm for Bi-KFS is provided in Algorithm 1. This strategy effectively propagates temporal context and suppresses noise in challenging dynamic scenes. Experiments show notable gains in both segmentation accuracy and temporal consistency for VCOS tasks.

## 4 Experiments

### 4.1 Datasets and Metrics

Details of the datasets, evaluation metrics, and implementation are provided in Appendix A.

### 4.2 Comparison with SOTA methods

**Quantitative Results.** As shown in Table 1, CamoTracer outperforms all previous methods across all metrics on both MoCA-Mask and CAD2016, including methods based on images and videos. Compared to the best method without using SAM, TMNet, CamoTracer improves $S_\alpha$ and mIoU by 0.06 and 0.176 on MoCA-Mask, corresponding to relative gains of 8.1% and 42.2%, respectively.

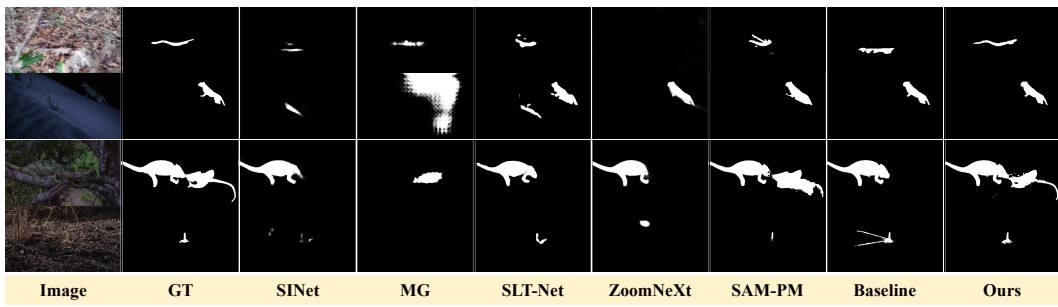

| Image | GT | SINet | MG | SLT-Net | ZoomNeXt | SAM-PM | Baseline | Ours |

Figure 3: Comparison of segmentation results between the SOTA methods, the baseline, and our proposed CamoTracer. The baseline used here is LISA with SAM2.

Table 2: Component ablation results on MoCA-TE dataset.

| Setting | SGA | SAP | Bi-KFS | $S_\alpha \uparrow$ | $F_\beta^w \uparrow$ | $E_\phi \uparrow$ | $\mathcal{M} \downarrow$ | $mDice \uparrow$ | $mIoU \uparrow$ |
|---|---|---|---|---|---|---|---|---|---|
| Baseline | | | | 0.653 | 0.359 | 0.710 | 0.039 | 0.384 | 0.325 |
| + SGA | ✓ | | | 0.735 | 0.527 | 0.790 | 0.017 | 0.546 | 0.478 |
| + SAP | | ✓ | | 0.735 | 0.524 | 0.767 | 0.012 | 0.532 | 0.475 |
| + SGA & SAP | ✓ | ✓ | | 0.771 | 0.606 | 0.822 | 0.009 | 0.614 | 0.541 |
| All | ✓ | ✓ | ✓ | **0.800** | **0.665** | **0.878** | **0.006** | **0.674** | **0.593** |

Table 3: Ablation study for different inference strategies on MoCA-TE dataset.

| Setting | $S_\alpha \uparrow$ | $F_\beta^w \uparrow$ | $E_\phi \uparrow$ | $\mathcal{M} \downarrow$ | $mDice \uparrow$ | $mIoU \uparrow$ |
|---|---|---|---|---|---|---|
| Forward | 0.771 | 0.606 | 0.822 | 0.009 | 0.614 | 0.541 |
| Backward | 0.746 | 0.557 | 0.817 | 0.007 | 0.570 | 0.498 |
| Bi-KFS | **0.800** | **0.665** | **0.878** | **0.006** | **0.674** | **0.593** |

Benefiting from the powerful segmentation capability of SAM2, CamoSAM2 surpasses all prior methods. Our CamoTracer further addresses the ambiguity of appearance and noise in motion estimation inherent in SAM-based methods, achieving additional improvements of 0.035 in $S_\alpha$ and 0.051 in mIoU on MoCA-Mask, corresponding to relative gains of 4.6% and 9.4%. On CAD2016, CamoTracer achieves increases of 0.056 in $S_\alpha$ and 0.121 in mIoU, representing improvements of 7.2% and 22.2%. These results demonstrate the superiority of our method.

**Qualitative Results.** As shown in the Fig. 3, we present a visual comparison of the segmentation results produced by our method and other methods. We use LISA with SAM2 we implemented as the baseline, which relies solely on text prompts and employs memory attention for tracking. In the first row, our method effectively alleviates visual ambiguity, successfully distinguishing camouflaged objects from the background. In the second row, the text prompts provided by the MLLM introduce shape priors, benefiting both the baseline and our method. The third row demonstrates our method's ability to perform fine-grained segmentation of multiple objects. In the fourth row, our method is capable of segmenting small objects that other models fail to detect.

## 4.3 Ablation Studies

We performed thorough ablation studies to validate our improvements, examining the contributions of each module, the effects of various prompt designs, and the influence of keyframe selection strategies.

**Modules.** As shown in Table 2, building upon the baseline, our proposed SGA module improves $S_\alpha$ and mIoU by 0.082 and 0.153, corresponding to relative gains of 12.6% and 47.1%, respectively, highlighting the importance of aligning visual and semantic representations. Meanwhile, the SAP module fully leverages the strong capabilities of SAM2 by providing robust prompt guidance, resulting in improvements of 12.6% in $S_\alpha$ and 46.2% in mIoU. When both modules are used together, they yield a combined gain of 0.118 in $S_\alpha$ and 0.216 in mIoU, corresponding to relative improvements of 18.1% and 66.5%. In addition, by incorporating the training-free Bi-KFS to enhance long-term temporal propagation and prediction consistency, we achieve further improvements. Compared to the variant without Bi-KFS, $S_\alpha$ and mIoU increase by 3.8% and 9.6%, respectively. Relative to the baseline, the gains reach 22.5% in $S_\alpha$ and 82.5% in mIoU.

**Different Prompts.** To validate the effectiveness of the three different prompt types and the memory attention module that functions as historical frame prompting, we conduct an ablation study as shown in Table 4. Each prompt positively contributes to the overall segmentation performance. For instance, removing the text prompt results in a decrease of 0.033 in $S_\alpha$ and 0.065 in mIoU, demonstrating the importance of semantic understanding. Similarly, removing the point prompt and mask prompt leads

Table 4: Ablation study for different prompts on MoCA-TE dataset.

| Mask | Point | Text | Memory | $S_\alpha \uparrow$ | $F_\beta^w \uparrow$ | $E_\phi \uparrow$ | $\mathcal{M} \downarrow$ | $mDice \uparrow$ | $mIoU \uparrow$ |
|---|---|---|---|---|---|---|---|---|---|
| ✓ | ✗ | ✗ | ✓ | 0.751 | 0.568 | 0.841 | 0.012 | 0.577 | 0.510 |
| ✗ | ✓ | ✗ | ✓ | 0.726 | 0.525 | 0.752 | 0.026 | 0.534 | 0.475 |
| ✗ | ✗ | ✓ | ✓ | 0.735 | 0.527 | 0.790 | 0.017 | 0.546 | 0.478 |
| ✗ | ✓ | ✓ | ✓ | 0.733 | 0.525 | 0.881 | 0.012 | 0.531 | 0.473 |
| ✓ | ✗ | ✓ | ✓ | 0.756 | 0.583 | 0.808 | 0.010 | 0.587 | 0.518 |
| ✓ | ✓ | ✗ | ✓ | 0.738 | 0.529 | 0.823 | 0.014 | 0.541 | 0.476 |
| ✓ | ✓ | ✓ | ✗ | 0.723 | 0.505 | 0.804 | 0.012 | 0.525 | 0.447 |
| ✓ | ✓ | ✓ | ✓ | **0.771** | **0.606** | **0.822** | **0.009** | **0.614** | **0.541** |

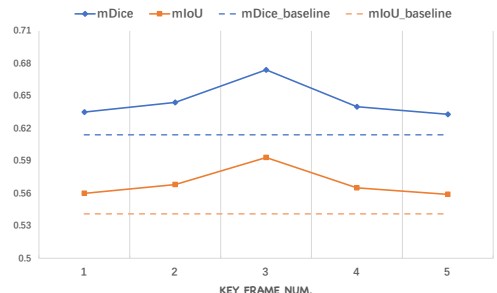

Figure 4: Impact of keyframe number.

to decreases in $S_\alpha$ by 0.015 and 0.038, and in mIoU by 0.023 and 0.068, respectively, highlighting the importance of explicit spatial cues for accurate segmentation. Moreover, when the memory attention is removed, the performance drops significantly. Specifically, $S_\alpha$ decreases by 0.048 and mIoU by 0.094, confirming the module's crucial role in maintaining consistent object tracking across frames.

**Different Inference Strategies.** As shown in Table 3, when video frames are processed in their natural forward order during a single inference, the model achieves an $S_\alpha$ score of 0.771. In contrast, processing the video in reverse order results in a slightly lower $S_\alpha$ of 0.746. However, for videos where camouflaged objects are difficult to detect in the early frames, utilizing contextual information from later frames can lead to more accurate segmentation, as shown in Fig. 6. Our proposed Bi-KFS strategy combines the advantages of both approaches by leveraging bi-directional contextual information during inference and mitigating the impact of inaccurate segmentation on subsequent frames. As a result, it achieves an improved $S_\alpha$ score of 0.800.

Additionally, we conduct experiments to analyze the effect of the number of key frames used in Bi-KFS, as shown in Fig. 4. The results show that using too few key frames leads to insufficient contextual information, whereas using too many introduces noise. Optimal performance is achieved when the number of key frames, $K$, is set to 3. Notably, the Bi-KFS strategy consistently outperforms both the forward-only (baseline in the figure) and backward-only inference strategies, regardless of the number of key frames.

### 4.4 VISUALIZATION

**Mask prompt and Point Prompt.** We show the mask prompt and point prompt generated by our method in Fig. 5. Compared to the baseline that relies solely on text prompts, our mask prompt successfully captures the chameleon on the right, enabling accurate segmentation of multiple objects.

**The effectiveness of Bi-KFS.** We present segmentation results produced by three different inference strategies in Fig. 6. The forward strategy suffers from inter-frame discontinuities, where objects segmented in one frame may disappear in subsequent frames. In contrast, the backward strategy propagates predictions from clearer frames to more ambiguous ones, resulting in improved accuracy. Our proposed Bi-KFS combines both directions, leveraging bidirectional contextual information to effectively mitigate inter-frame discontinuities.

### 5 CONCLUSION

We present CamoTracer, the first VCOS framework that integrates MLLMs with SAM2 to achieve fully automated, semantics-rich prompt generation. By introducing the Semantic-Guided Adapter and Semantic-Aware Prompter, our approach bridges the gap between visual and language modalities, enabling robust segmentation in camouflaged scenes. Additionally, our Bi-directional Keyframe Selection strategy enhances temporal consistency through memory-guided propagation. Extensive experiments demonstrate that CamoTracer surpasses previous SOTA by a large margin, marking a promising step toward general-purpose, LLM-driven video segmentation in challenging camouflage scenarios. In future work, we plan to extend our framework to broader video segmentation tasks in open-world settings.

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

## A  DATASETS AND METRICS

**Datasets.** We evaluate our method on two widely used video camouflaged object detection (VCOD) benchmarks: MoCA-Mask (Cheng et al., 2022b) and CAD2016 (Bideau & Learned-Miller, 2016b). MoCA-Mask is derived from the MoCA dataset and provides dense pixel-level annotations of camouflaged animals in dynamic natural scenes. It consists of 87 video sequences, including 19,313 frames from 71 sequences for training and 3,626 frames from 16 sequences for testing, with annotations every fifth frame. CAD2016 is a compact evaluation-only dataset composed of 9 short clips (836 frames in total), with manually annotated segmentation masks also sampled every five frames.

**Evaluation Metrics.** We adopt six standard metrics for quantitative evaluation: S-measure ($S_\alpha$), which evaluates structural similarity; Weighted F-measure ($F_\beta^w$), balancing precision and recall with spatial weighting; Enhanced-alignment measure ($E_\phi$), assessing both region-aware and pixel-level alignment; Mean Absolute Error ($\mathcal{M}$), measuring average pixel-wise deviation; mean Dice (mDice),

and mean Intersection-over-Union (mIoU), both of which quantify region overlap. Higher scores of $S_\alpha$, $F_\beta^w$, $E_\phi$, mDice, and mIoU, along with a lower $\mathcal{M}$, indicate better segmentation performance.

**Implementation Details.** We train the model using two NVIDIA 24G 3090 GPUs with a distributed training script based on DeepSpeed (Rasley et al., 2020). We use the AdamW (Loshchilov & Hutter, 2017) optimizer with the learning rate and weight decay set to 3e-4 and 0, respectively. We adopt WarmupDecayLR as the learning rate scheduler, with the warmup iterations set to 100. The weights of the text generation loss ($\lambda_{txt}$), the mask loss ($\lambda_{mask}$) and the coarse mask loss ($\lambda_{coarse}$) are all set to 1.0. The weights of the BCE loss ($\lambda_{bce}$) and the DICE loss ($\lambda_{dice}$) are set to 2.0 and 0.5, respectively. Following Cheng et al. (2022a), we use the training set of COD10K (3,040 images) (Fan et al., 2021b) and the training set of MoCA-Mask (19,313 frames) and evaluate on the MoCA-Mask test set, as well as on the entire CAD2016 dataset. We choose the hiera-L version of SAM2 and LISA-7B (Lai et al., 2024) in all experiments. We train CamoTracer for 10 epochs with a per-device batch size of 2.

# B   VISUALIZATION RESULTS

Below are the visualizations of our experimental results.

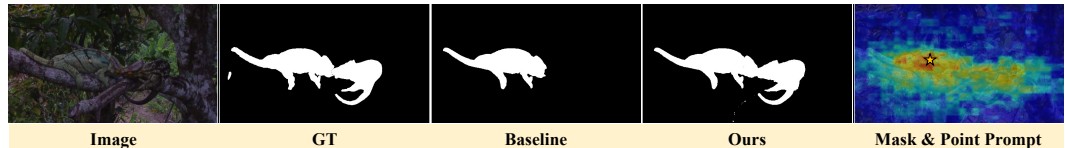

Figure 5: Visualization of the mask prompt and point prompt generated in our method.

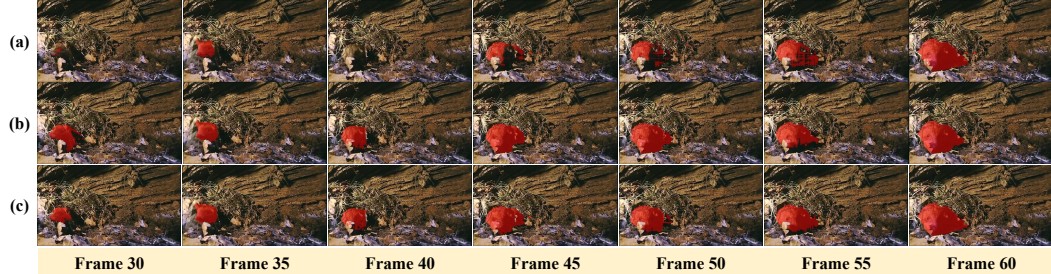

Figure 6: Visualization of segmentation results under different inference strategies: (a) forward-only; (b) backward-only; (c) Bi-KFS.

# C   COMPARISON WITH RECENT MLLMS

Table 5 presents a comprehensive comparison between our proposed CamoTracer and several recent multimodal large language models (MLLMs), including PixelLM, LISA, LISA++, and VideoLISA, evaluated on two challenging VCOS benchmarks: MoCA-Mask and CAD2016. While generic MLLMs demonstrate certain capabilities in multimodal understanding, their performance on camouflaged object segmentation remains suboptimal. This is primarily due to the unique challenges in camouflage scenarios, such as high background-foreground similarity and low object saliency, which are not explicitly addressed in generic MLLM training. As the results show, our method outperforms all baselines across all six metrics on both datasets, achieving notable gains in structure-aware measures ($S_\alpha$, $F_\beta^w$, $E_\phi$) as well as region-aware scores (mDice, mIoU). In particular, on the MoCA-Mask dataset, CamoTracer surpasses the strongest baseline (VideoLISA) by large margins in $F_\beta^w$ (0.665 vs. 0.273), mDice (0.674 vs. 0.309), and mIoU (0.593 vs. 0.246). A similar trend is observed on CAD2016, where our model achieves state-of-the-art performance with $E_\phi = 0.900$, mDice = 0.750, and mIoU = 0.664. These consistent improvements validate the importance of domain-specific

Table 5: Comparison with recent MLLMs on MoCA-Mask and CAD2016 datasets. The best results are **bolded**. Our CamoTracer outperforms all baselines across six metrics, demonstrating its superior capability in segmenting camouflaged objects with higher accuracy, robustness, and generalizability.

| Method | MoCA-Mask | | | | | | CAD2016 | | | | | |
|---|---|---|---|---|---|---|---|---|---|---|---|---|
| | $S_\alpha \uparrow$ | $F_\beta^w \uparrow$ | $E_\phi \uparrow$ | $\mathcal{M} \downarrow$ | $mDice \uparrow$ | $mIoU \uparrow$ | $S_\alpha \uparrow$ | $F_\beta^w \uparrow$ | $E_\phi \uparrow$ | $\mathcal{M} \downarrow$ | $mDice \uparrow$ | $mIoU \uparrow$ |
| PixelLM (Ren et al., 2024b) | 0.476 | 0.113 | 0.504 | 0.135 | 0.135 | 0.104 | 0.552 | 0.368 | 0.594 | 0.218 | 0.391 | 0.314 |
| LISA (Lai et al., 2024) | 0.552 | 0.179 | 0.627 | 0.037 | 0.209 | 0.158 | 0.650 | 0.424 | 0.742 | 0.037 | 0.418 | 0.335 |
| LISA++ (Yang et al., 2023) | 0.509 | 0.156 | 0.633 | 0.099 | 0.171 | 0.132 | 0.604 | 0.385 | 0.786 | 0.053 | 0.401 | 0.313 |
| VideoLISA (Bai et al., 2024) | 0.557 | 0.273 | 0.621 | 0.133 | 0.309 | 0.246 | 0.696 | 0.530 | 0.798 | 0.054 | 0.569 | 0.474 |
| CamoTracer (Ours) | **0.800** | **0.665** | **0.878** | **0.006** | **0.674** | **0.593** | **0.830** | **0.745** | **0.900** | **0.014** | **0.750** | **0.664** |

architectural enhancements and temporal modeling, as well as the effectiveness of task-oriented fine-tuning. In contrast to generic MLLMs, CamoTracer is specifically tailored to the demands of video camouflaged object segmentation, leading to significant performance gains and more reliable predictions in complex scenes.

## D  PARAMETER AND TRADE-OFF ANALYSIS

Table 6 compares the tuning parameter and segmentation performance of different methods on the MoCA-Mask and CAD2016 VCOS benchmarks. Our model, CamoTracer, achieves the best performance across all evaluation metrics, with only a marginal increase in parameters (291.38M) compared to the LISA+SAM2 baseline (288.26M). Despite this small overhead of only 1.1% in parameter size, our model yields substantial improvements: a relative gain of 85.2% in $F_\beta^w$ and 82.5% in mIoU on MoCA-Mask, and similar improvements on CAD2016. This strong boost originates from our specifically designed lightweight modules that enhance temporal coherence and semantic alignment without significantly increasing computational burden.

Compared to SLT-Net and TSP-SAM, CamoTracer consistently outperforms even under stricter resource constraints. Importantly, we are the first to introduce a *multimodal large language model* (MLLM) tailored for *video camouflaged object segmentation (VCOS)*, which integrates visual-language reasoning via task-specific instruction tuning. To ensure parameter efficiency, we adopt LoRA-based tuning, enabling effective multimodal alignment with minimal trainable overhead. These design choices allow CamoTracer to push the state of the art in VCOS while remaining computationally tractable for practical deployment.

Beyond accuracy, we also report inference efficiency on RTX3090 (batch=1). CamoTracer runs at 2.63 FPS with ~3131 GFLOPs and ~15.5 GB memory usage, which is comparable to FSPNet (2.94 FPS) and TSP-SAM (2.53 FPS) while delivering much higher accuracy (+9.4% mIoU over CamoSAM2 and +52.8% mIoU over TSP-SAM). This confirms that our design achieves a favorable accuracy–efficiency trade-off.

Table 6: Tuning parameters and segmentation performance of different methods. The best results are **bolded**.

| Method | Tuning Params (M) | MoCA-Mask | | | | | | CAD2016 | | | | | |
|---|---|---|---|---|---|---|---|---|---|---|---|---|---|
| | | $S_\alpha \uparrow$ | $F_\beta^w \uparrow$ | $E_\phi \uparrow$ | $\mathcal{M} \downarrow$ | $mDice \uparrow$ | $mIoU \uparrow$ | $S_\alpha \uparrow$ | $F_\beta^w \uparrow$ | $E_\phi \uparrow$ | $\mathcal{M} \downarrow$ | $mDice \uparrow$ | $mIoU \uparrow$ |
| FSPNet (Huang et al., 2023) | 274.24 | 0.594 | 0.182 | 0.608 | 0.044 | 0.238 | 0.167 | 0.539 | 0.220 | 0.553 | 0.145 | 0.309 | 0.212 |
| SLT-Net (Cheng et al., 2022a) | **82.38** | 0.656 | 0.357 | 0.785 | 0.021 | 0.387 | 0.310 | 0.679 | 0.420 | 0.805 | 0.033 | 0.445 | 0.342 |
| TSP-SAM (Hui et al., 2024b) | 89.78 | 0.689 | 0.444 | 0.808 | 0.008 | 0.458 | 0.388 | 0.704 | 0.524 | 0.912 | 0.028 | 0.543 | 0.438 |
| CamoSAM2 (Zhang et al., 2025a) | 95.5 | 0.765 | 0.607 | 0.848 | 0.007 | 0.620 | 0.542 | 0.774 | 0.652 | 0.852 | 0.018 | 0.647 | 0.543 |
| Baseline (LISA w/ SAM2) | 288.26 | 0.653 | 0.359 | 0.710 | 0.039 | 0.384 | 0.325 | 0.805 | 0.677 | 0.885 | 0.017 | 0.693 | 0.601 |
| CamoTracer (Ours) | 291.38 | **0.800** | **0.665** | **0.878** | **0.006** | **0.674** | **0.593** | **0.830** | **0.745** | **0.900** | **0.014** | **0.750** | **0.664** |

## E  GENERALIZATION EVALUATION

To assess the generalizability of our framework beyond camouflaged scenes, we evaluated CamoTracer on DAVIS-2016 (Perazzi et al., 2016), a benchmark for generic video object segmentation, under the unsupervised setting. As shown in Table 7, our method achieves a J&F score of 90.3, outperforming recent SOTA methods such as GFA (Song et al., 2024) (88.2) and (Cho et al., 2024) (87.6). This

Table 7: Performance Comparison with SOTA methods over DAVIS2016.

| Method | Pub. | J&F | J | F |
|---|---|---|---|---|
| DTTT (Liu et al., 2024) | CVPR 2024 | 87.2 | 85.8 | 88.5 |
| DPA (Cho et al., 2024) | CVPR 2024 | 87.6 | 86.8 | 88.4 |
| GSA (Lee et al., 2024) | CVPR 2024 | 87.7 | 87.0 | 88.4 |
| GFA (Song et al., 2024) | AAAI 2024 | 88.2 | 87.4 | 88.9 |
| TMO (Cho et al., 2025) | Arxiv 2025 | 88.2 | 88.0 | 88.3 |
| CamoTracer (ours) | - | **90.3** | **90.1** | **90.4** |

result demonstrates that the proposed prompt-free video segmentation framework possesses strong transferability.

## F  ALGORITHM OVERVIEW FOR BI-KFS STRATEGY

To clarify the implementation of our proposed keyframe selection strategy, we provide the pseudo code of the Memory-Guided Bi-Directional Keyframe Selection (Bi-KFS) in Algorithm 1. This method leverages bi-directional segmentation predictions to estimate frame-wise consistency and selects keyframes based on both confidence and temporal coherence. By combining consistency-aware ranking with high-confidence filtering, the strategy ensures reliable keyframe selection under challenging camouflaged scenarios. This not only improves temporal robustness but also facilitates memory-efficient processing for downstream segmentation.

---

**Algorithm 1:** Memory-Guided Bi-Directional Keyframe Selection

---

**Input:** Video frames $\{I_1, I_2, \ldots, I_T\}$; Segmentation model $\mathcal{F}$; Confidence threshold $\tau$; Number of keyframes $K$;

**Output:** Keyframe set $\mathcal{K}$;

**Step 1: Bi-directional Inference**;

**for** $t = 1, 2, \ldots, T$ **do**
    $M_t^{\text{fwd}}, \text{Conf}_t^{\text{fwd}} \leftarrow \mathcal{F}(I_{1:t})$;
    $M_t^{\text{bwd}}, \text{Conf}_t^{\text{bwd}} \leftarrow \mathcal{F}(I_{T:t})$;
    $S_t \leftarrow \text{IoU}(M_t^{\text{fwd}}, M_t^{\text{bwd}})$;

**Step 2: Select Candidates by Consistency**;

Sort frames by $S_t$ in descending order;

Select top-$K$ frames $\{t_1, t_2, \ldots, t_K\}$;

**Step 3: Select High-Confidence Frames**;

**for** $t = 1, 2, \ldots, T$ **do**
    **if** $\max(Conf_t^{fwd}, Conf_t^{bwd}) > \tau$ **then**
        Add $t$ to high-confidence set $\mathcal{H}$;

**Step 4: Finalize Keyframes**;

**for** *each* $t \in \{t_1, t_2, \ldots, t_K\} \cup \mathcal{H}$ **do**
    **if** $Conf_t^{fwd} > Conf_t^{bwd}$ **then**
        $\mathcal{K} \leftarrow \mathcal{K} \cup \{(I_t, M_t^{\text{fwd}})\}$;
    **else**
        $\mathcal{K} \leftarrow \mathcal{K} \cup \{(I_t, M_t^{\text{bwd}})\}$;

**Return:** Keyframe set $\mathcal{K}$;

---

## G  QUALITATIVE COMPARISON OF VIDEO SEGMENTATION

To further validate the effectiveness of our method, we present qualitative comparisons on challenging video sequences. As illustrated in the supplementary video (demo.mp4), our model consistently produces high-quality masks that accurately delineate camouflaged objects across frames, even under

severe boundary ambiguity, foreground occlusion, camera shake, morphological similarity, dynamic occlusion, color homogeneity and structural resemblance. Compared to existing methods, which often suffer from mask fragmentation or temporal inconsistency, CamoTracer exhibits robust temporal coherence and precise boundary localization. This visual evidence underscores the benefits of our multimodal guidance and temporal modeling modules, and highlights the superiority of our approach in real-world camouflaged scenarios.

## H  LIMITATIONS AND FUTURE WORK

While CamoTracer demonstrates strong performance in VCOS, it still faces several practical challenges. First, the bi-directional inference strategy introduces additional computational overhead compared to one-way propagation, although it significantly improves temporal consistency and is practical for short-to-medium video sequences. Second, the quality of semantic prompts generated by the multimodal model may occasionally be suboptimal in complex or cluttered scenarios, potentially leading to imperfect guidance. Finally, due to memory constraints, our training primarily focuses on short video clips, which may limit temporal modeling in extremely long sequences. These issues are not fundamental flaws but represent areas where further optimization could yield broader applicability and efficiency.

In future work, we plan to explore lightweight alternatives for bi-directional inference, such as adaptive keyframe scheduling or early-exit mechanisms, to reduce computational cost while preserving accuracy. To improve semantic alignment, we aim to incorporate temporally-aware language grounding or refinement modules that better adapt to dynamic visual scenes. Moreover, we intend to extend CamoTracer's temporal scope by integrating memory-efficient recurrent architectures or hierarchical temporal sampling, enabling robust performance in longer and more diverse video sequences.

