# OpenReview forum: "Tracing the Hidden: Segment Anything in Camouflaged Videos via Prompt-Free Multimodal LLM Guidance"
_ICLR.cc/2026/Conference — ICLR 2026 Conference Withdrawn Submission_

### Official Review · Reviewer_PeNY · 2025-10-26

**Soundness:** 3
**Presentation:** 3
**Contribution:** 3
**Rating:** 4
**Confidence:** 5

**Summary:**

This paper presents a novel framework CamoTracer for video camouflaged object segmentation (VCOS) designed to automate prompt generation for the SAM 2 model.
The proposed method leverages a MLLM to generate diverse, intervention-free prompts (point, mask, and text).
Key components include a Semantic-Guided Adapter to align MLLM and SAM 2 features, and a Semantic-Aware Prompter to convert MLLM outputs into diverse prompts.
Additionally, a memory-guided bi-directional keyframe selection strategy is employed to enhance temporal consistency.
The method achieves better performance on the VCOS benchmark.

**Strengths:**

1. The paper proposes a novel and timely framework that integrates the MLLM with the SAM 2 model for VCOS, using an MLLM to generate diverse, multimodal prompts automatically (point, mask, text).
2. The introduction of the Semantic-Guided Adapter and the Semantic-Aware Prompter provides a good way to bridge MLLM semantics with SAM 2's visual features and prompt embeddings.
3. The paper proposes a memory-guided Bi-directional Keyframe Selection strategy to improve temporal propagation and reliability, addressing specific VCOS challenges like occlusion and irregular motion.

**Weaknesses:**

1. The distinction between "OURS" and "OURS*" in Fig. 1 is unclear. This should be explicitly clarified in the caption.
2. Regarding the mechanism of human visual reasoning and temporal memory (lines 091-094), is this an empirical summary, or is it supported by specific experiments? Citations to supporting literature are required here for this claim to be reliable.
3. In Tab. 1, following current community practices, the missing metric results should be completed if the original papers provided models, code, or predictions. For instance, to my knowledge, EMIP has publicly available code.
4. In Tab. 1, the authors should add details on computational complexity (e.g., FLOPs/latency) and default inference resolution for all compared methods. Furthermore, the backbone models and key components used by each method should be specified to ensure a clearer and fairer comparison.
5. The specific architecture of the baseline model used in the ablation study is inadequately described. While it seems to be a LISA + SAM2 combination, more details on how they are integrated are required. Concurrently, in Tab. 2, why does the introduction of SGA and SAP to the baseline result in such a large performance boost? The performance on MoCA-TE even surpasses many carefully-designed methods like TMNet, IMEX, TSP-SAM, and EMIP.
6. In the ablation study, the mechanisms of action for SGA and SAP are not clearly analyzed or validated.
    1. SGA appears to be a simple feature fusion component integrating CLIP features into SAM 2's workflow. Why does this yield such a significant performance gain?
    2. The performance improvement from SAP seems to derive largely from its effect on the Prompt Embedding.
    3. Although the authors provide a comparison of different prompt combinations (Tab. 4), a clear baseline for this specific ablation is missing. Why does the "+Point" variant show such large differences in metrics compared to other variants? For example, while its mIoU and $F^{\omega}_{\beta}$ are only 0.003 and 0.002 different from "+Text", other metrics show large gaps, up to 0.038 ($E_{\phi}$). Furthermore, the best $E_{\phi}$ score comes from the "+Point+Text" variant, not the final full model. The extremely high $E_{\phi}$ for "+Mask" is also questionable and casts some doubt on the reliability of these results.
7. The performance results in Fig. 4 appear to be symmetrical around $K=3$ (i.e., $K=1$ results match $K=5$, and $K=2$ results match $K=4$). Why does this perfect symmetry occur?
8. The term "Prompt-Free" is misleading. The paper repeatedly claims the framework is "prompt-free", but while it is "human-prompt-free", it is not entirely "prompt-free". The entire MLLM guidance process relies on a fixed, manually-designed text instruction: `<IMAGE> Please segment the camouflaged object in this image.`. This is a task-specific, hard-coded prompt. The model's performance likely depends on the specific wording of this instruction, yet no sensitivity analysis is provided.

**Questions:**

See Weaknesses.

---

### Official Review · Reviewer_i6W1 · 2025-10-30

**Soundness:** 2
**Presentation:** 3
**Contribution:** 2
**Rating:** 4
**Confidence:** 5

**Summary:**

This paper proposes CamoTracer for video camouflaged object segmentation (VCOS). The key idea is to leverage multimodal large language models (MLLMs) to generate text, mask, and point prompts in a fully automatic manner, guiding the SAM2 model for segmenting camouflaged objects without human annotation. The method introduces two main modules: the Semantic-Guided Adapter (SGA) and the Semantic-Aware Prompter (SAP), along with a Bi-directional Keyframe Selection (Bi-KFS) strategy for enhancing temporal consistency. The authors conduct experiments on MoCA-Mask and CAD2016 datasets.

**Strengths:**

1. The motivation is good. Exploring the potential of SAM2 for camouflaged object segmentation is a well-motivated direction. Camouflage is an especially challenging setting that can highlight the model’s limitations and strengths.
2. The performance of the proposed method is good.
3. The paper is easy to read.

**Weaknesses:**

1. The writing quality is relatively weak.
2. The core innovation is not tightly coupled with the camouflage task itself. If the authors aim to explore the potential of SAM2 under challenging perception scenarios, it would be more convincing to include additional context-dependent concept (CD concept) tasks such as salient object detection, shadow detection, or medical lesion segmentation, which share similar characteristics with camouflage.
3. The technical novelty is limited. The idea of using MLLMs to generate prompts has already been explored in multiple recent works. In this paper, the novelty mainly lies in system integration and application rather than in conceptual or algorithmic design. The proposed Semantic-Guided Adapter and Semantic-Aware Prompter lack clear technical insight or architectural novelty.
4. Table 1 shows good performance, but since the method leverages powerful foundation models, it is necessary to report inference speed, FLOPs to evaluate the trade-off between accuracy and efficiency. Without such comparisons, the performance gain may come at a substantial computational cost.
5. The baseline design is inappropriate. Since the core motivation is to eliminate SAM2’s dependence on manual prompts, the baseline should be SAM2 with dataset-driven prompts, such as using ground truth masks from previous frames. The current comparison does not clarify whether the rich prompts generated by MLLMs can match or exceed the performance of manually prompted SAM2 in the VCOS setting.

**Questions:**

See the weaknesses.

---

### Official Review · Reviewer_ZRZ9 · 2025-10-30

**Soundness:** 2
**Presentation:** 3
**Contribution:** 2
**Rating:** 4
**Confidence:** 4

**Summary:**

This paper presents CamoTracer, a framework that integrates multimodal large language models (MLLMs) with SAM2 to achieve fully automatic, prompt-free camouflaged video segmentation. By introducing a Semantic-Guided Adapter, a Semantic-Aware Prompter, and a bi-directional keyframe selection strategy, it enables self-generated semantic prompts for temporally consistent and robust segmentation under challenging camouflage conditions.

**Strengths:**

1.The paper tackles a challenging and underexplored task — camouflaged video segmentation — and proposes a novel prompt-free yet prompt-rich paradigm.

2. The integration of MLLMs with SAM2 via the Semantic-Guided Adapter and Semantic-Aware Prompter is technically sound, enabling automatic multimodal prompt generation.

3. Extensive experiments on multiple benchmarks demonstrate clear state-of-the-art performance and strong generalisation across diverse camouflage scenarios.

**Weaknesses:**

1. Computational complexity. The framework combines SAM2, CLIP, and an MLLM with multiple cross-attention and adapter layers, making it computationally heavy. Inference latency and deployment feasibility are not discussed.

2. Supervision dependency. Although described as prompt-free, the method still requires dense supervision for training. The high annotation cost of camouflaged videos limits scalability and somewhat contradicts the “no human involvement” claim.

3. Limited conceptual novelty. The idea of automatic prompt generation overlaps with prior works like GenSAM[1], MMCPF[2] and ProMaC[3]. The paper should better clarify how its semantic-temporal prompting goes beyond existing image-level promptable segmentation frameworks.

[1] Hu, Jian, et al. "Relax image-specific prompt requirement in sam: A single generic prompt for segmenting camouflaged objects." Proceedings of the AAAI Conference on Artificial Intelligence. Vol. 38. No. 11. 2024.

[2] Tang, Lv, et al. "Chain of visual perception: Harnessing multimodal large language models for zero-shot camouflaged object detection." Proceedings of the 32nd ACM international conference on multimedia. 2024.

[3] Hu, Jian, et al. "Leveraging hallucinations to reduce manual prompt dependency in promptable segmentation." Advances in Neural Information Processing Systems 37 (2024): 107171-107197.

**Questions:**

1. The proposed Bi-KFS relies on forward–backward IoU consistency for keyframe selection. How does it handle failure cases when the initial frame prediction is inaccurate or under fast motion/occlusion?

2. The generated prompts are claimed to be “semantic and adaptive,” but their interpretability and actual influence on segmentation remain unclear. Could the authors provide more analysis or visualization to support this claim?

---

### Official Review · Reviewer_57iY · 2025-11-01

**Soundness:** 2
**Presentation:** 3
**Contribution:** 3
**Rating:** 4
**Confidence:** 3

**Summary:**

To segment camouflaged objects SAM framework is used often. This
framework requires prompts. Here, authors present a way to use LLMs to
create such prompts without requiring human interaction. To this end,
authors fuse CLIP and SAM through an adapter and a prompter that
converts text prompts into SAM-style prompts. The framework is
designed to segmented objects in videos. To that end, a key-frame
selection strategy is also devised.

**Strengths:**

+ to the best of my knowledge, this is one of the first works that
  integrates guidance through language to SAM prompting for segmenting
  camouflaged objects in videos.
+ generation of prompts from text input coming from LLM is well
  done.
+ the results shown in Table 1 are pretty impressive.
+ the ablation studies present value of different components.

**Weaknesses:**

- the description of the method can improve.
  + In Equation 1, it is unclear what $x_{img}^{(t)}$ and $x_{txt}$
    stand for. Is the former the output of the CLIP for each patch
    representation or is it the actual image considering that authors
    use a MLLM? The same representation is later used in Eqn. 2 as the
    input to the SAM2 encoder, which increases the confusion
    further. Is the latter the text instruction given above or
    something else. Is it the same for all the frames in a video?
  + Where is $f_{clip}$ defined?
- while I understand the reasoning behind the prompter, I do not
  understand the intuition behind the adapter. Authors state that the
  SAM2 features capture fine-grained details while the CLIP provides
  contextual information. To me it seems like the adapter is not
  integral to the model. It would also work without it and thus the
  adapter seems like a heuristic solution to increase accuracy.
- the description of the experimental details could
  improve. Specifically, how are ground truth text $y_{txt}$'s defined?
- Including a language model to generate prompts for SAM led to best
  results across the board. Given such high results, I think it is
  very important to provide a very sound argument why the model
  outperforms all other SOTA methods in this way. Such an
  argumentation is missing in this paper. It is especially the case
  when one considers that the prompt fed to the LLM is rather
  simplistic and not optimized at all. The article is written as a
  "recipe" and I think that does not make the proposed method shine.

**Questions:**

+ please improve the description of the method as well as the
  experiments. The lack of clarity is hurting your article.
+ It would be interesting to run the model with oracle prompts and
  evaluate how well the LLM approach is able to generate similar
  results.

---

### Note · Authors · 2025-12-04

I have read and agree with the venue's withdrawal policy on behalf of myself and my co-authors.